# Dialectical Behaviour Therapy Improves Emotion Dysregulation Mainly in Binge Eating Disorder and Bulimia Nervosa: A Systematic Review and Meta-Analysis

**DOI:** 10.3390/jpm11090931

**Published:** 2021-09-18

**Authors:** Natalia Rozakou-Soumalia, Ştefana Dârvariu, Jan Magnus Sjögren

**Affiliations:** Psychiatric Centre Ballerup, Eating Disorder Research Unit, 2750 Copenhagen, Denmark; natalia.rozakou.soumalia@regionh.dk (N.R.-S.); stefana.darvariu@regionh.dk (S.D.)

**Keywords:** emotion regulation, eating disorders, dialectical behaviour therapy, binge eating disorder, bulimia nervosa

## Abstract

Emotion dysregulation is a transdiagnostic phenomenon in Eating Disorders (ED), and Dialectical Behaviour Therapy (DBT) (which was developed for reducing dysregulated emotions in personality disorders) has been employed in patients with ED. This systematic review and meta-analysis investigated whether the effect of DBT was stronger on emotion dysregulation, general psychopathology, and Body Mass Index (BMI) in participants with ED, when compared to a control group (active therapy and waitlist). Eleven studies were identified in a systematic search in accordance with PRISMA guidelines. Most studies included participants with Binge Eating Disorder (BED) (*n* = 8), some with Bulimia Nervosa (BN) (*n* = 3), and only one with Anorexia Nervosa (AN). The pooled effect of DBT indicated a greater improvement in Emotion Regulation (ER) (g = −0.69, *p* = 0.01), depressive symptoms (g = −0.33, *p* < 0.00001), ED psychopathology (MD = −0.90, *p* = 0.005), Objective Binge Episodes (OBE) (MD = −0.27, *p* = 0.003), and BMI (MD = −1.93, *p* = 0.01) compared to the control group. No improvement was detected in eating ER following DBT (*p* = 0.41). DBT demonstrated greater efficacy compared with the control group in improving emotion dysregulation, ED psychopathology, and BMI in ED. The limitations included the small number of studies and high variability.

## 1. Introduction

Eating disorders (ED) are defined as bio-psycho-social disorders, likely caused by a combination of genetic, psychological, and sociocultural factors. The Diagnostic and Statistical Manual of Mental Disorders (DSM) describes three major ED, namely Anorexia Nervosa (AN), Bulimia Nervosa (BN), and Binge-Eating Disorder (BED), while also including Other Specified Feeding and Eating Disorders (OSFED), previously termed Eating Disorder Not Otherwise Specified (EDNOS) [1]. In the last 20 years, an increasing trend has been observed in the prevalence of ED, from 3.5% to 7.8%.

These disorders are characterised by high mortality rates as well, which indicates an escalating health threat [2]. AN is responsible for 5.1 deaths/1000 person-years (1.3 deaths due to suicide), BN for 1.74 deaths/1000 person-years and BED together with OSFED for 3.31 deaths/1000 person-years [3]. This is further underscored by the poor treatment outcome, where only 64% of AN and 45% of BN achieved full remission, and around 20% and 23% endured chronic AN and BN, respectively. The mean follow up in these studies was 25 years for AN and 30 years for BN [4,5]. Patients with BED and OSFED presented the highest remission rate (70%), while 10% of BED patients still had the disorder at the 5-year follow up [6].

Emotion dysregulation is a transdiagnostic phenomenon in ED [7], with Emotion Regulation (ER) defined as all the processes that influence and control the emotions that individuals experience. The influences on emotion regulation capacity can be external (typically family and friends), or internal, neurophysiological processes (e.g., the development of cognitive skills) [8,9]. In ED, emotion dysregulation is characterised by enhanced emotional responses, which are further amplified by reduced emotional awareness and adaptability [7].

In other words, the emotional dimension is strongly associated with ED psychopathology, which raises the question of its impact in treating ED [10,11]. The most commonly investigated treatments in ED are Cognitive Behaviour Therapy (CBT) and Interpersonal Psychotherapy (IPT). The former achieves reduced emotional distress and problematic behaviours through addressing automatic maladaptive thought patterns and challenging their validity.

In contrast, IPT aims to strengthen the individual’s social support, improve their interpersonal skills, and decrease emotional distress through addressing the social and interaction difficulties frequently observed in ED patients, which may stem from a crucial interpersonal problem in their lives [12,13,14,15]. While emotion dysregulation is part of these treatments, none of these specifically focus on this component. Therapies are more concerned either with elements of cognition, such as selective attention or cognitive control in CBT or with aspects of interpersonal capacity and functioning in IPT.

DBT (Dialectical Behavioural Therapy) is based on biosocial theory, developed by Linehan (1993), which explains how biological vulnerability together with an invalidating environment result in emotion dysregulation [16]. The latter affects psychological and social functioning (e.g., the experience of the sense of self, behaviour control, interpersonal relationships, etc.). This theory highlights the need for patients to obtain new skills that can be used in the individual’s environment to neutralize these deficits [17].

According to Linehan (1993), the treatment must strengthen the patient’s skills and motivate the patient to employ the skills and apply them in all relevant contexts. The strategies and skills utilized in DBT are categorized as those that encourage acceptance of reality and those that promote change. For instance, the process of learning how to change emotions or interpersonal relationships is the focus of change skills, while acceptance of situations and emotional pain is the aim of mindfulness and distress tolerance [18].

Since the biosocial theory framework also applies in ED patients, DBT was suggested as a treatment in these disorders. DBT’s focus on emotion dysregulation as well as its initial development for patients with multiple problems (Axis I and Axis II comorbidities are commonly observed in ED) demonstrate DBT’s eligibility in treating ED [19]. In the past few years, DBT has been gaining empirical support for the treatment of all typical ED, with the evidence being more extensive in the case of BED and BN [20,21,22]. Previous studies have investigated the effect of DBT in ED patients with and without comorbidity [23] as well as adaptations and implementation of DBT [24,25,26,27]. Most studies found an encouraging rate of binge abstinence (from 29–89%) and reduction in the frequency of ED symptoms [20].

Despite awareness that emotion dysregulation is a transdiagnostic phenomena in ED, the state of knowledge on the efficacy of DBT in influencing Emotion Regulation (ER) in ED is currently unknown. Since ER may have a substantial impact on the treatment outcomes in all types of ED, this is important to investigate. Previous studies only focused on summarizing the literature on psychological interventions for ED and included DBT as one of these interventions [18,19] without specifying its individual effects.

We therefore undertook the first systematic review and meta-analysis of the effect of DBT in ED and specifically studied the effect of DBT on ER in comparison to a control group. In addition, we also investigated the effect on general psychopathology and Body Mass Index (BMI) as secondary aims. The ambition was to provide a current state of evidence for the effect of DBT by only including RCT with high-quality of evidence in order to provide guidance for clinical decisions and directions for future research on the use of DBT in ED.

## 2. Materials and Methods

### 2.1. Search Strategy and Study Selection

A systematic review and a meta-analysis were carried out in accordance with the Preferred Reporting Items for Systematic Reviews and Meta-Analyses (PRISMA) guidelines and the Cochrane Handbook for Systematic Reviews [28,29]. The protocol was registered in the International Prospective Register of Systematic Reviews, PROSPERO (CRD42021223633). The electronic databases Pubmed, Embase (Ovid), PsychInfo (EBSCO), and ClinicalTrials.gov were thoroughly searched on the 7th of December 2020. Further, the reference lists of the included articles were assessed for the identification of additional studies.

The keywords included the major, as well as less common ED and spelling variations of DBT. A detailed description of search terms and strategy can be viewed in Appendix A. The search results were imported in Endnote for duplicate removal, a step that was reiterated in Rayyan [30,31]. Assessment of the retrieved publications was carried out by two authors (S.D. and N.R.-S.), with input from a third author (J.M.S.) in the event of disagreements.

### 2.2. Eligibility Criteria

The inclusion criteria for studies were: (1) participants diagnosed with an ED or atypical ED, (2) studies investigating the effect of DBT, (3) studies including ER as an outcome, (4) studies with a control group (active therapy or wait list), (5) clinical trials (randomised and non-randomised), (6) written in English and (7) published after 2000. This date cut-off was selected as the state of knowledge regarding ED, especially BED, has been maturing over the last 20 years and due to absence of classification and knowledge about the application of DBT in ED before that time period. Although a few papers may have been published before 2000, these were still considered when found through the identified publications.

In terms of exclusion criteria, these were: (1) reviews, meta-analyses, or editorials, as original publications were sought, to allow an unbiased analysis, (2) studies that lacked a control, which was considered a requirement for determining the efficacy of DBT, (3) participants below 12 years old, since children are yet to develop the cognitive skills to identify and report cognitions and emotional states, (4) participants with substance abuse, (5) pregnant participants, both (4) and (5) are considered as confounding states where emotions may be affected differently than in ED and thus not representative of the general ED population.

### 2.3. Outcomes and Data Extraction

The primary outcome of this study was Emotion Regulation (ER) represented in several ways as follows: (1) primary ER (one Outcome Measure Instrument [OMI] per study), (2) overall ER (all ER-related OMI per study), also divided into general ER (OMI depicting general presence or regulation of negative affect (DERS, NMRS, EDI-3 emotion dysregulation subscale, and PANAS)) and eating ER (OMI depicting desire to eat in response to negative affect (EES and DEBQ)). Depressive symptoms (OMI indicating depressive symptoms (BDI, EES depression subscale and PANAS-NA)) were considered reflective of ER.

The secondary outcomes were ED psychopathology, as well as BMI. The former was investigated as (1) the frequency of key behavioural features of ED, such as restraint, eating concern, weight concern, shape concern, and the presence of binge eating behaviours (Eating Disorder Examination-Questionnaire (EDE-Q) and Binge Eating Scale (BES)) and (2) frequency of binge-eating, which was measured as Objective Binge Episodes (OBE). For the systematic review, essential information—including OMI—was extracted and organized in Table 1 and Table 2. Any missing data that were relevant to the study were requested from the corresponding author of the specific publication.

### 2.4. Quality Assessment and Risk of Bias in the Included Studies

The risk of bias in the included randomised and non-randomised studies was assessed according to the Cochrane Collaboration’s Tool Risk of Bias 1 (RoB 1) and Risk of Bias in Non-randomised Studies-I (ROBINS-I), respectively. The assessment was performed by S.D. and N.R.-S. independently, both at the study and outcome level across all bias domains. Any discrepancies were discussed until a consensus was reached. The key domains for the subjective and objective outcomes were identified and evaluated across randomised studies based on their relative importance in the current review.

The domains that were evaluated in the randomised papers were the following: random sequence generation, allocation concealment, participants, personnel and outcome assessor blinding and incomplete outcome data. Regarding the non-randomised study, the domains that were assessed were: bias due to confounding, deviations from intended interventions, missing data, bias in selection of the study participants, classification of interventions and measurement of outcomes. The studies were rated as low, intermediate, or high quality based on the level of risk of bias at each key bias domain, which would correspond to one or more domains at high risk, one or more domains at unclear risk, or all domains at low risk of bias, respectively.

### 2.5. Statistical Analysis

Due to the absence of comparison analyses and standard deviations in the included studies, Welch’s two-sample *t*-tests and SD imputations were conducted. The pooled effects of the outcomes of interest were estimated in Review Manager 5.4.1.—where a random-effects meta-analysis model was used—in which the mean changes from baseline to post-treatment were included [32]. The effect size was presented as the weighted mean differences (MD) or Hedges’ g—when outcomes were assessed by different instruments across papers—with 95% confidence intervals (CI). The cut-off points were applied according to the multicomponent empirically-based guidelines for rehabilitation treatment by Kinney et al. (2020) [33].

As such, effect sizes up to 0.31 were considered as small, between 0.31–0.55 as medium, and over 0.55 as large. When more relevant measures were available per study, all of them were included. To avert the statistical interpretation of the same studies as independent ones, and overestimation of the effects, adjustments were carried out. This correction was achieved through dividing the sample size by the number of times the study was introduced in the analysis.

A negative value demonstrated an effect favouring the experimental group, whereas a positive indicated an effect favouring the control (active therapy and wait list). For some outcome measures, a positive coefficient was an indication of improvement (e.g., NMRS, PANAS positive, etc.). Thus, in these cases, the signs were reversed where DBT had a greater absolute value compared with the control.

The degree of heterogeneity across studies was depicted by I^2^ statistics, which was evaluated based on the guidelines created by Higgins et al. Based on these, the I^2^ can indicate low (25–50%), moderate (50–75%), or high (>75%) heterogeneity [34]. Due to differences in the binge frequency measurements between studies, binge rates were calculated as follows: Binge days/episodes were divided by 28, the number of days the measure was reflective of. In one paper, the values were first multiplied by 4 and then divided by 28, as the measure was recorded weekly [35].

Subgroup analyses based on comparison group (active therapy vs. wait list), gender (male vs. female), ED type (BED vs. miscellaneous) and ER outcome (primary studies-ER as primary outcome vs. secondary studies-ER as a secondary outcome) were performed to investigate possible differences between groups in the magnitude of DBT’s effect. Meta-regression analyses were also conducted subsequently to explore sources of heterogeneity and to indicate the possible influence of different predictor variables, such as country (US vs. Non-US), BMI, age, sample size, and ED type.

## 3. Results

### 3.1. Study Selection

The literature search revealed a total of 535 eligible articles. Among those, 27 were reviewed for eligibility in full text, according to the inclusion and exclusion criteria. Two studies were included as one—seeing that one paper presented the post-treatment results and the other one reported the follow-up findings [36]—and 11 were ultimately included in the systematic review and meta-analysis (see Figure 1).

### 3.2. Study Characteristics

The 11 studies included 669 participants of ages ranging from 18–65 years, assigned either to a DBT group or a control group. (In some studies, the latter consisted of a waitlist group (six studies) whereas others comprised an evidence-based active therapy form (five studies). The active therapy comparisons were CBT and CBT+, Behavioural Treatment Plan (BTP), Active Comparison Group Therapy (ACGT), and Supportive Group Therapy (SGT).

Among the studies, 10 were RCTs (one of which breached the randomisation technique occasionally when it was convenient) [37], and only one was a non-randomised controlled clinical trial [36]. The studies were published in the time period between 2001 and 2020, and interventions lasted from 10 up to 24 weeks with either one or two sessions per week. Some studies included a follow-up period which ranged from 3 months to a year, with one notably longer, which lasted 6 years [36].

Most studies were conducted in the USA (*n* = 6) and the remaining were from the Netherlands, Iran, Spain, and Canada. The preponderance of participants had BED, with one study examining BED and loss of control. Only two papers included participants diagnosed with BN and one investigated participants with AN, BN, and OSFED. The majority of the studies considered exclusively female participants, and the few studies including males still had predominantly women participants (more than 85%) [35,37,38,39,40]. More details on the characteristics of the included studies can be seen in Table 1 and Table 2.

**Table 1 jpm-11-00931-t001:** Characteristics of the included studies (DBT vs. WL).

First Author, Year	Study Design	Sample Size (*n*)	Population Demographics	ED Type and Diagnostic Manual	Treatment	Key Findings Post-Treatment
Dastan et al., 2019 [41]	RCT Follow-up: -	Total: 40 DBT: 20 WL: 20	Age (range): 18–50 Females: 100% Nationality: Iranian	ED: BED Criteria: DSM-IV Assessment: SCID-I	Intervention: DBT vs. WL (no intervention) Duration: 20 weeks (2 h session/week)	1. Emotion regulation -EES global, anger/frustration, anxiety subscales: significantly higher improvements in DBT -EES depression: no significant difference 2. BMI: significantly higher reduction in DBT
Hill, 2007 [42]	RCT Follow-up: -	Total: 32 DBT: 18 WL: 14	Age (SD): 22 (6.3) Females: 100% Nationality: American	ED: BN (*n* = 26) and sub-clinical BN (*n* = 6) Criteria: DSM-IV Assessment: SCID-I	Intervention: DBT vs. WL (6-week delayed intervention) Duration: 12 weeks (1 h session/week: the length of the first 6 sessions was increased to 90 min)	1. Emotion regulation -EES, NMRS, PANAS-NA: no significant differences -PANAS-PA, BDI-II: significantly higher improvements in DBT 2. ED psychopathology -EDE-Q global, restraint, shape, eating concern: significantly higher improvements in DBT -Weight concern: no significant differences -OBE: significantly greater reductions in DBT
Masson et al., 2013 [39]	RCT Follow-up: 6 months	Total: 60 DBT: 30 WL: 30	Age (SD): 42.8 (10.5) Females: 88.3% Males: 11.7% Nationality: Canadian	ED: BED Criteria: DSM-IV Assessment: SCID-I and EDE	Intervention: DBT-guided self-help vs. WL (no intervention) Duration: 13 weeks (6 biweekly 20-min support phone calls)	1. Emotion regulation -DERS: significantly greater improvements in DBT 2. ED psychopathology -EDE-Q all subscales, except eating concern: significantly higher improvements in DBT -OBE: significantly greater reductions in DBT -EDQLS: significantly greater improvements in DBT
Rahmani et al., 2018 [43]	RCT Follow-up: -	Total: 60 DBT: 30 WL: 30	Age (SD): 30.5 (7.5) Females: 100% Nationality: Iranian	ED: BED Criteria: DSM-IV-TR Assessment: SCID-DSM-IV-TR	Intervention: DBT vs. WL (offered treatment at the end of study) Duration: 10 weeks (2 h session/twice a week)	1. Emotion regulation -DERS: significantly greater improvements in DBT 2. ED psychopathology -BES: significantly greater reductions in DBT 3. BMI: significantly greater reductions in DBT
Safer et al., 2001 [44]	RCT Follow-up: -	Total: 29 DBT: 14 WL: 15	Age (SD): 34 (11) Females: 100% Nationality: American	ED: BN (80.6%) and sub-clinical BN (19.4%) Criteria: DSM-IV Assessment: EDE	Intervention: DBT vs. WL (offered treatment at the end of study) Duration: 20 weeks	1. Emotion regulation -EES global and subscales, PANAS-NA, NMRS, BDI: significantly greater improvements in DBT -PANAS-PA: no significant differences 2. ED psychopathology -OBE: significantly greater reductions in DBT
Telch et al., 2001 [45]	RCT Follow-up: -	Total: 44 DBT: 22 WL: 22	Age (SD): 50 (6.1) Females: 100% Nationality: American	ED: BED Criteria:DSM-IV Assessment: SCID-I and SCID-II	Intervention: DBT vs. WL (no intervention) Duration: 20 weeks (2 h session/week)	1. Emotion regulation -EES global, anxiety and depression: no significant differences -EES anger: greater improvements in DBT (borderline significant) -PANAS, NMRS, BDI: no significant differences 2. ED psychopathology -OBE, BES: significantly greater reductions in DBT -EDE-Q weight, shape, eating concern: significantly higher improvements in DBT -EDE-Q global and restraint concern: no significant differences

**Table 2 jpm-11-00931-t002:** Characteristics of the included studies (DBT vs. AT).

First Author, Year	Study Design	Sample Size (*n*)	Population Demographics	ED Type and Diagnostic Manual	Treatment	Key Findings Post-Treatment
Adler, 2008 [35]	RCT Follow-up: 18 weeks	Total: 17 DBT: 8 BTP: 9	Age (SD): 49.4 (11.4) Females: 88.2% Males: 11.8% Nationality: American	ED: BED (subthreshold: *n* = 4) Criteria: DSM-IV Assessment: EDE	Intervention: DBT + Alli (weight loss drug) vs. BTP + Alli Duration: 12 weeks (2 h session/week)	1. Emotion regulation -EES, BDI: no significant differences 2. ED psychopathology -OBE: no significant differences -BES: significantly greater reductions in DBT
Hoffman, 2006 [38]	RCT Follow-up: -	Total: 101 DBT: 50 SGT: 51	Age (SD): 51.6 (11.2) Females: 85.1% Males: 14.9% Nationality: American	ED: BED Criteria: DSM-IV Assessment: SCID-I and SCID-II	Intervention: DBT vs. SGT Duration: 20 weeks (2 h session/week)	1. Emotion regulation -DERS: no significant difference -BDI: significantly greater reductions in DBT 2. ED psychopathology -OBE: significantly greater reductions 3. BMI: no significant differences in DBT
Lammers et al., 2020 [37]	Quasi-randomised control trial Follow-up: 6 months	Total: 74 DBT: 41 CBT+: 33	Age (SD): 37.3 (11.8) Females: 89.2% Males: 10.8% Nationality: Dutch	ED: BED Criteria: DSM-V Assessment: DEBQ and SCID	Intervention: DBT-BED vs. CBT+ Duration: 20 weeks (2 h session/week)	1. Emotion regulation -DEBQ, EDI-3 (emotion dysregulation), BDI-II: no significant difference 2. ED psychopathology -EDE-Q, EDI-3 (self-esteem), SCL-90: no significant differences -OBE: significantly greater reductions in CBT+ 3. BMI: no significant differences in DBT
Navarro-Haro et al., 2020 [36]	Non-randomised control trial Follow-up: 4 and 6 years	Total: 109 DBT: 64 TAU-CBT: 45	Age (SD): 27.3 (8.1) Females: 100% Nationality: Spanish Comorbidity: BPD	ED: BN, AN or EDNOS Criteria: DSM-IV Assessment: SCID-I	Intervention: DBT vs. TAU-CBT Duration: 6 months (2 h session/week)	1. Emotion regulation -EES: significantly greater improvements in TAU-CBT -ERQ, BDI-II, PANAS-PA: no significant difference -PANAS-NA: borderline significant
Safer et al., 2010 [40]	RCT Follow-up: 3, 6 and 12 months	Total: 101 DBT: 50 ACGT: 51	Age (SD): 52.2 (10.6) Females: 86% Males: 14% Nationality: American	ED: BED Criteria: DSM-IV Assessment: SCID-I	Intervention: DBT-BED vs. ACGT Duration: 21 weeks (2 h session/week)	1. Emotion regulation -NMRS, EES, PANAS, DERS: no significant difference -BDI: significantly greater reductions in DBT 2. ED psychopathology -EDE-Q global and eating concern: significantly greater improvements in DBT -Restraint, weight and shape concern: no significant improvements in DBT -OBE: significantly reduced in DBT 3. BMI: no significant differences

Abbreviations for Table 1 and Table 2: Active Comparison Group Therapy (ACGT), Active Therapy (AT), Anorexia Nervosa (AN), Beck Depression Inventory (BDI), Behavioural Treatment Plan (BTP), Binge Eating Disorder (BED), Binge Eating Scale (BES), Body Mass Index (BMI), Borderline Personality Disorder (BPD), Bulimia Nervosa (BN), Cognitive Behaviour Therapy (CBT), Diagnostic and Statistical Manual of Mental Disorders (DSM), Dialectical Behaviour Therapy (DBT), Difficulties in Emotion Regulation Scale (DERS), Dutch Eating Behaviour Questionnaire (DEBQ), Eating Disorder(s) (ED), Eating Disorder Examination (Questionnaire) (EDE(-Q)), Eating Disorder Inventory (EDI-3, emotion dysregulation subscale), Eating Disorder Quality of Life Scale (EDQLS), Eating Disorders not Otherwise Specified (EDNOS), Emotional Eating Scale (EES), Emotion Regulation Questionnaire (ERQ), Negative Mood Regulation Scale (NMRS), Objective Binge Episodes (OBE), Positive and Negative Affect Schedule (PANAS), Positive affect (PA), Negative affect (NA), Randomised Control Trial (RCT), Structural Clinical Interview for DSM (SCID), Standard Deviation (SD), Supportive Group Therapy (SGT), Symptom Checklist (SCL-90), Treatment as Usual (TAU), and Waitlist (WL).

### 3.3. Quality Assessment

In total, the overall quality of the current review was rated as intermediate with three studies at low risk of bias [37,39,41], four at unclear risk [35,40,43,44], and three at high risk [38,42,45]. The domains at the highest risk of bias were randomisation and allocation concealment, due to a lack of explicit mentioning. For the non-randomised study, all key domains were at moderate risk, except for deviations from the intended interventions and missing data, which were at low risk. Overall, the quality of the non-randomised study was moderate and in line with the total quality of the current review. The figures are available in Appendix A.

### 3.4. Effectiveness of DBT

#### 3.4.1. Emotion Regulation (ER)

The analysis including only primary outcomes revealed a significant effect of DBT on ER (g = −0.69, *p* = 0.01), accompanied by a high heterogeneity (I^2^ = 90%), as well as a wide confidence interval (95% CI: [−1.22, −0.16]) (Figure 2). The analysis including all ER measures available showed a significant pooled effect (*p* < 0.001) in favour of DBT. The standardised mean difference was g = −0.46 with a 95% confidence interval of [−0.67, −0.26] and presented high heterogeneity (I^2^ = 81%) (Table A1, Appendix B). After adjusting, a small decrease in the standardised mean difference was observed; however, the analysis still yielded a significant effect (*p* = 0.001) in favour of DBT. The heterogeneity decreased substantially (I^2^ = 53%) (Figure 3).

Distinguishing between the types of affect regulation, DBT had a significant effect only on general ER (*p* < 0.01). The effect was five-times greater than in eating ER, while the heterogeneity and the confidence interval were also wider (g = −0.70, 95% CI: [−1.23, −0.18], *p* = 0.009, I^2^ = 89%, g = −0.15, 95% CI: [−0.51, 0.21], *p* = 0.41, I^2^ = 69%) (Table A1, Appendix B). After adjustment, the effect sizes were unchanged (Table A2, Appendix B). The overall effect of DBT on depressive symptoms favoured DBT, with a significant effect size of g = −0.33 (95% CI: [−0.45, −0.20], *p* < 0.00001, I^2^ = 9%). Following adjustment, the effect remained significant, whilst the heterogeneity diminished to zero (see Figure 4).

##### Subgroup Analyses

Overall, there was a significant subgroup difference between waitlist and active therapy studies, as observed in overall, general and eating Emotion Regulation (ER) (Appendix A). The trend displayed higher effects of DBT in studies that compared to a wait list group, as opposed to those comparing to active therapy studies, and the difference was a three-fold and ten-fold increase, respectively. Heterogeneity and confidence intervals were also notably larger in wait list studies relative to active therapy. In terms of depressive symptoms, there was no significant difference between wait list and active therapy studies. Subgroup differences were identified in overall ER between studies that exclusively investigated females and those that also included males (Appendix A).

While both presented a significant effect, this was four-times higher in the females-only subgroup. No significant subgroup difference was identified in general ER and depressive symptoms. The females-only group showed greater effects, especially in general ER where the effect size was over six-times higher. Heterogeneity was generally low in the mixed gender group and high in the females-only group, except for depressive symptoms, where both were low. The CI followed similar patterns. There were no subgroup differences identified in overall and general ER, where the effect sizes were significant in both subgroups and larger in BED-only studies (Appendix A).

Heterogeneity was low only in the mixed groups, while the CI were generally wide in both subgroups. With regards to depressive symptoms, the effects were also significant in both groups and larger in the mixed subgroup, which presented a wide CI and zero heterogeneity. The BED-only group also had low heterogeneity and was accompanied by a narrower CI.

There was no subgroup difference between primary and secondary ER studies in overall and general ER, as both showed significant effect sizes (Appendix A). They were higher in studies primarily focusing on ER, which also presented wider CI and heterogeneity. The latter was absent in secondary studies. In the case of depressive symptoms, the effect was greater and only significant in secondary studies, but the subgroup test was not significant. Heterogeneity was 0 in both groups, while the CI was only wide in primary studies.

#### 3.4.2. ED Psychopathology

##### Severity of Symptoms

The overall effect of DBT on the severity of symptoms favoured DBT, with g = −0.90 (*p* = 0.002) and a wide confidence interval (95% CI: [−1.45, −0.34]) (Table A1, Appendix B). After adjustment, the effect remained significant (Figure 5).

##### Objective Binge Episodes (OBE)

The total effect of DBT on OBE followed a negative direction (MD = −0.27, 95% CI: [−0.45, −0.09], *p* = 0.003), along with a heterogeneity of 85%, indicating a greater reduction of OBE in DBT (Figure 6).

##### Subgroup Analyses

The subgroup analyses showed higher effects of DBT on ED psychopathology in studies with a waitlist group. However, no significant differences were detected between the waitlist and active therapy groups in OBE analysis (*p* = 0.12, I^2^ = 59%) (Appendix A). Studies with exclusively female participants showed larger effect sizes compared to mixed gender studies, namely nine-times higher in symptoms and three-times greater in OBE.

The tests for subgroup differences yielded a non-significant result (Appendix A). Subgroup analysis by ED type showed a greater effect on OBE in the miscellaneous ED group compared to the BED group. Both groups were characterised by high heterogeneity (I^2^ = 78% and I^2^ = 91%, respectively). However, no significant difference was detected between the groups (Appendix A).

#### 3.4.3. Body Mass Index (BMI)

The effect of DBT on BMI was significant (*p* = 0.01) with a MD = −1.93, 95% CI: [−3.42, −0.44] and presented moderate heterogeneity I^2^ = 32% (Figure 7).

##### Subgroup Analyses

A significantly greater effect was seen in the exclusively female group compared to the mixed gender group, while both groups presented wide confidence intervals. However, the difference was not reflected in the subgroup test, which yielded non-significant results between the groups (*p* = 0.27) (Appendix A).

### 3.5. Meta-Regression Analysis

When investigating heterogeneity in overall Emotion Regulation (ER), no single explanatory variable significantly accounted for the between-study variance. When testing the interaction between predictor variables, the highest proportion of explained variance was detected in the model based on BMI and age (R^2^ = 91%, 95% CI: [0.02, 0.10], 95% CI: [−0.05, 0.00], respectively). Only BMI showed a significant influence (*p* < 0.01) (Appendix A). No other combinations of predictors explained a higher proportion of the variance. The full meta-regression analysis and bubble plots are available in Appendix A.

### 3.6. Publication Bias

The funnel plot for overall ER and depressive symptoms showed no evidence of asymmetric distribution of studies, as the points of the plot mostly scattered evenly on both sides of the reference line, suggesting a low risk of publication bias (Appendix A). With regards to variation of the sampling distribution, there was a gap in lower-powered articles, as all publications included in this review tended to be aggregated in the upper part of the plot.

## 4. Discussion

The aim of this study was to investigate primarily the effect of DBT on Emotion Regulation (ER) in ED and secondarily its effect on ED psychopathology and BMI as compared to a reference treatment or wait list group. The systematic review and meta-analysis of the current literature indicated an improvement following DBT on all outcomes, with clearer effects on general psychopathology and BMI.

### 4.1. Emotion Regulation (ER)

In the meta-analysis, DBT showed a moderate to large improvement in the ER of ED participants (primary model: g = −0.56; overall model: g = −0.42). The current results were mainly applicable in BED, as sample sizes for BN and especially AN were considerably lower. Therefore, the findings have very limited reliability in the context of other ED, as the investigation of DBT in BN and AN is in its infancy. Currently, the effects of DBT on reducing difficulties in ER have only been documented in a systematic review, which investigated a mixed population [46]. Their findings indicated a possible effect of DBT on ER, while emphasising the inability to draw clear conclusions on the topic due to sparse and inconsistent data. In contrast, the current study examined a more homogeneous population, while still lacking consistency in the use of active controls, as well as DBT variations.

As defined in the methods, distinguishing between the two types of affect regulation (general versus eating ER), a significant influence of DBT was only observed in general Emotion Regulation (ER). The large effect size indicates that DBT may be effective in improving the general regulation of emotions, rather than directly modulating disordered eating in reaction to negative affect (g = −0.70). However, these mechanisms may be dependent on each other, since emotional eating has previously been associated with difficulties in the specific dimensions of ER [47,48].

Emotional eating induced by negative affective states—especially depression—was specifically associated with increased difficulties of engaging in goal-directed behaviour, controlling impulse and accepting emotional reactions. Eating triggered by negative affect was also associated with insufficient access to emotion regulation strategies [49]. Negative emotional eating may act as a precursor for experiencing disordered eating symptoms, which emphasises the relevance of targeting this, along with poor emotion regulation in treatment. Therefore, future studies could focus on creating adaptations of DBT with increased focus on these particular dimensions of general ER, as well as assessing emotion regulation and emotional eating as two separate constructs [50].

Longer follow-up periods may also be necessary to observe larger changes in eating ER, since the current literature search has revealed that this is a rather uncommon finding with only one study presenting a considerable follow-up period of 6 years [36].

The smallest effect of DBT was observed in the treatment of depressive symptoms (g = −0.29). Even though the improvement was small, it can be of great importance, due to the fact that a decrease in depressive symptoms and behaviours may reduce or even prevent self-injury behaviours, as well as suicide attempts and deaths in ED patients [51]. The latter is relevant in both AN and BN, where suicide is one of the most common causes of death [52,53]. Apart from the use of ER skills, implementation of interpersonal effectiveness skills that patients are taught in DBT may alternatively explain this result. The focus of this module is on insufficient psychosocial functioning—an essential element in the process of treating depression and depressive symptoms—typically not used by other therapies [54].

### 4.2. ED Psychopathology

The included studies showed consistent DBT-induced improvements in binge frequency and severity, along with reductions in overall intensity of ED symptoms. This was confirmed by the largest effect seen in the meta-analysis results in severity of symptoms (g = −0.83). The subscales of shape, weight and eating concern were improved in the majority of the studies, implying that DBT may have a specific effect on ED psychopathology. The restraint scores showed an improvement only in a few studies, causing inconsistencies in the interpretation. This could be explained by the dissimilarities between DBT interventions employed in the studies, as some may focus more on mindfulness, leading indirectly to a higher reduction in the restrictive attitude of the participants [40].

An improvement following DBT was also observed in studies that assessed quality of life, participants’ relationship with their body-self and appearance, as well as self-esteem. The mindfulness module taught in DBT may be responsible for improving body acceptance, by enhancing the present-moment experience and reducing the obsessive thoughts of body-appearance dissatisfaction [55]. Since increased severity of symptoms can have a serious impact on everyday life, reduction of ED psychopathology can lead to an increase in the quality of life in ED [56].

### 4.3. Body Mass Index (BMI)

When reviewing the existing literature, DBT was associated with no significant improvement in BMI in most of the included papers. In the majority of these studies, only post values were considered and used to assess the effect of DBT, which decreased the robustness of the analysis. Although the meta-analysis revealed a significant large effect on BMI (MD = −1.93), the findings cannot be extrapolated to all ED patients, since predominantly BED studies were used in this analysis.

This result was consistent with other research carried out in obese individuals who did not meet the criteria for BED, where BMI was significantly decreased after DBT [57]. Binge eating has been associated with emotional eating—especially in obese individuals—whether diagnosed with BED or not [58,59]. Therefore, BMI improvement is expected following DBT, due to an enhancement in patients’ ER skills and a reduction in binge eating cycles and the corresponding caloric intake. This highlights DBT’s potentially important role in reducing obesity rates and other diseases in later life, such as metabolic syndrome, which BED patients have been associated with [60,61,62,63].

### 4.4. Subgroup Analyses

DBT showed improvements in all outcomes when compared to wait list, whereas in active therapy studies superiority of DBT transpired only in depressive symptoms. The lack of an effect in the active therapy group may imply that DBT has similar effects to the other therapies. Previous studies have shown that CBT was equally beneficial to behavioural therapies in improving ED symptomatology [64,65]. However, this may be due to contrasting DBT to a mixture of therapies rather than exclusively to individual ones. Consequently, future trials should be conducted investigating differences between DBT and other common therapies in their potential to improve Emotion Regulation (ER). Increasing the level of singular evidence can facilitate reaching clearer conclusions through reviews and meta-analyses.

The role of gender on the psychotherapeutic outcomes of DBT may be considered, since higher effects were seen in studies exclusively consisting of females. Some studies propose that there are subtle differences in the male ED experience in terms of clinical presentation and underlying causes. Seeing that females have been the primary focus when developing treatment models for ED, males may require gender specific adaptations to existing treatment approaches, as well as to the diagnostic framework and measures of symptom severity [66,67]. In the current study, the group that also included males was still predominantly composed of women (∼85%), confirming the underrepresentation of the male population affected by ED. This can also be due to differences between men and women in terms of help-seeking attitudes, as well as the use of mental health services [68,69]. Nevertheless, the sample was mixed and not representative, and thus caution must be taken when interpreting these results.

A notable difference was observed in overall and general Emotion Regulation (ER), as DBT elicited double the effect in participants with BED. The latter is characterized by higher impulsivity, which is likely the driving factor underlying rapid avoidance of emotional experience and resolution [70]. On account of DBT’s key effects in impulsive behaviours, possible explanations could be acquisition of skills involved in distress tolerance, along with awareness of impulsive and emotion-driven behaviours [71,72]. In contrast, depressive symptoms and binge episodes showed larger effects in miscellaneous ED, groups mainly composed of BN. Since the sample size was much lower, adequately powered studies are required to confirm these ED specific differences in the effect of DBT.

Generally, effects were higher in studies that reported ER as a primary outcome than in those that included it as a secondary or exploratory outcome. Taken together with the considerable difference in heterogeneity between the corresponding subgroups (i.e., considerable heterogeneity in primary studies and negligible in secondary studies), these results suggest that the pooled estimate in the secondary studies subgroup may be closer to the true effect. Seeing that the result of an RCT is generally determined by its primary outcome, detection bias could represent a possible underlying reason for a much more pronounced effect [73].

### 4.5. Strengths and Limitations

This is the first systematic review on DBT in ED that includes a meta-analysis and the first study to provide effects of DBT on ER alongside effects on ED psychopathology and BMI. Moreover, exclusively including controlled trials allowed the comparison of the results with a reference group, i.e., excluding uncontrolled trials led to statistically stronger and more interpretable results [74]. In order to increase the robustness of the analysis and avoid excluding data that were not calculated or analysed in the original studies (e.g., SD), a set of statistical adaptations and calculations were conducted. This allowed the collection of more data, increasing the power of the analysis. Finally, instead of comparing differences in mean post-treatment values between groups, changes pre- and post-treatment were analysed. This ensured more robust results, as baseline differences were accounted for.

The current study presented a number of limitations worth reflecting upon. First, despite funnel plots indicating no risk of publication bias, the assessment was hypothetical since the figures are based on the non-adjusted analyses. Thus, the strength of the bias assessment was limited by the insufficient number of studies introduced in the analysis. Further, no test was conducted to examine asymmetry, and the plots were only inspected visually. The results were also limited by the moderate quality and considerable heterogeneity in the included studies.

#### Quality of the Included Studies and Heterogeneity

This study scored moderate in the quality assessment, indicating flaws in the methodologies of the included publications. The main shortcomings were found in random sequence generation and allocation concealment, as well as in the blinding of participants, personnel and outcome assessors. These are especially important because of the subjective nature of the outcomes, and thus emphasis should be placed on designing study methodologies that clearly distinguish and report these aspects. Another weakness in the included studies was the lack of consensus regarding outcome measurement instruments for Emotion Regulation (ER).

These varied considerably between the studies as different methods of measuring ER were used. This increases the difficulty of comparing and combining findings in a systematic review and meta-analysis [75]. While informative, indirect measures are more difficult to interpret, as the possibility of making cause and effect inferences becomes more challenging, an issue also highlighted in a previous review on DBT [20]. More direct measures of affect regulation, such as the Emotion Regulation Questionnaire (ERQ) and Difficulties in Emotion Regulation Scale (DERS) may represent changes in ER more accurately [76,77].

Moreover, heterogeneity was considered relatively high, indicating variability in the study designs and outcome measurements between the studies. In a further exploration of heterogeneity, the interaction between BMI and age explained the highest proportion of variance. A lower BMI was associated with a larger effect on Emotion Regulation (ER), while a younger age was associated with a smaller effect. This indicates a possible optimal weight, in addition to an age “window” where DBT may be more suitable. A greater effect with increasing age suggests that DBT may be more efficient in older populations, thus, reinforcing a potential association between emotional regulatory changes that occur with ageing and psychotherapy efficacy. In terms of ER, earlier research has supported that age is associated with the development of more stable affective experiences [78,79].

Nevertheless, developmental research in psychology has addressed different specific challenges that arise in psychotherapy with ageing (e.g., vocabulary used in therapy sessions). Integrating existing literature with the current results may indicate a demand for adapting DBT to fit the requirements of younger adults with ED.

### 4.6. Future Directions

Future investigations are necessary to validate the conclusions that are drawn from this study. First, the efficacy of DBT in emotion regulation should be examined in specific ED, rather than looking into all ED jointly. This may allow for future comparison between adapted forms of DBT and better determination of their effectiveness on the corresponding disorder. Secondly, as previously mentioned, only a few studies that included participants with BN and AN were available, suggesting that this area is still in its infancy. Therefore, further research in these two populations is pertinent to determine if there are any discrepancies between different types of ED regarding DBT’s efficacy on emotion dysregulation. Thirdly, it is recommended to focus on the male population (which is generally underrepresented in the included studies on eating disorders), as well as the adolescent population, since the existing studies consisted mainly of females, and no relevant study was found with adolescents.

## 5. Conclusions

The present results indicate a beneficial effect of DBT on improving emotion dysregulation in ED patients, including depressive symptoms, as well as ED psychopathology and BMI. However, there are not many studies consistently comparing DBT to active therapy for ED or investigating other ED than BED. Consequently, further studies are required to determine the effectiveness of DBT in patients with BN, AN, and OSFED, as well as research contrasting DBT to known ED therapies, such as CBT and IPT. Additionally, it would be pertinent to analyse individual factors (e.g., gender, age, and BMI) which can contribute to the development of ED specific interventions.

## Figures and Tables

**Figure 1 jpm-11-00931-f001:**
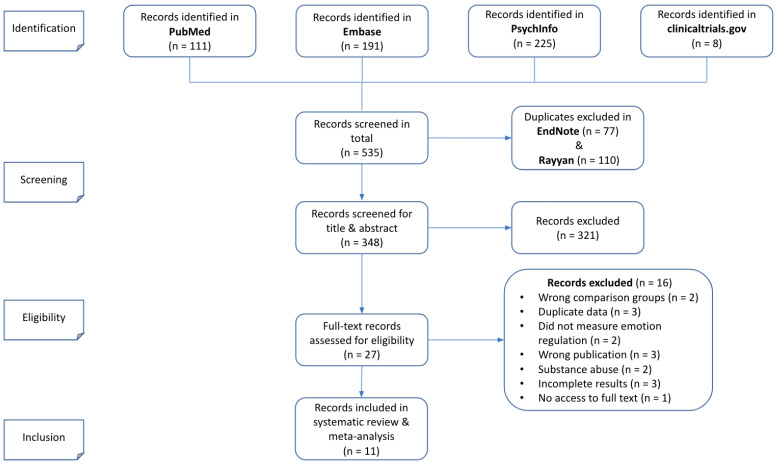
Prisma flow diagram of the current study.

**Figure 2 jpm-11-00931-f002:**
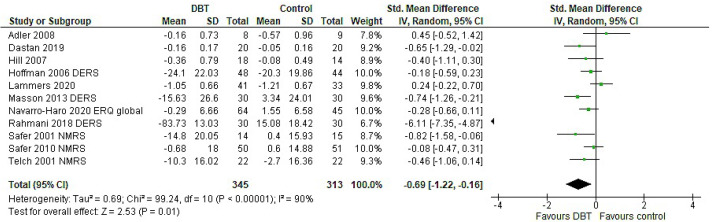
Forest plot: analysis of Emotion Regulation (ER) including one (primary) outcome per study.

**Figure 3 jpm-11-00931-f003:**
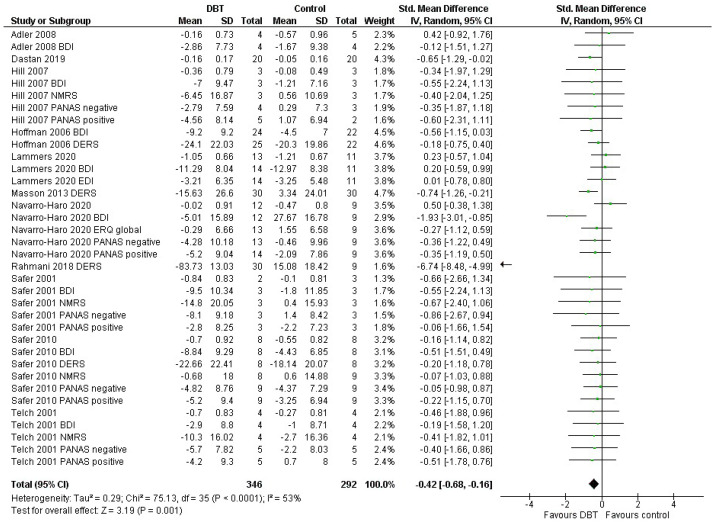
Forest plot: adjusted analysis of Emotion Regulation (ER) including all relevant outcomes per study.

**Figure 4 jpm-11-00931-f004:**
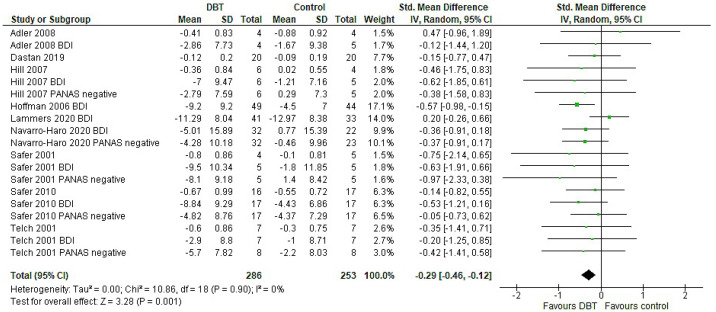
Forest plot: adjusted analysis of depressive symptoms including all relevant outcomes per study.

**Figure 5 jpm-11-00931-f005:**
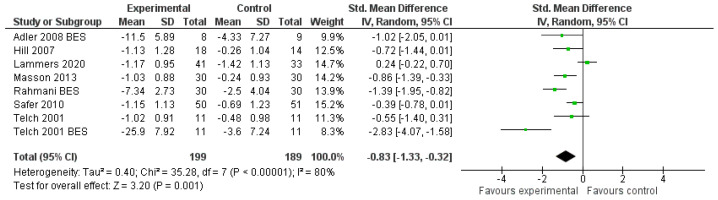
Forest plot: adjusted analysis of the severity of symptoms.

**Figure 6 jpm-11-00931-f006:**
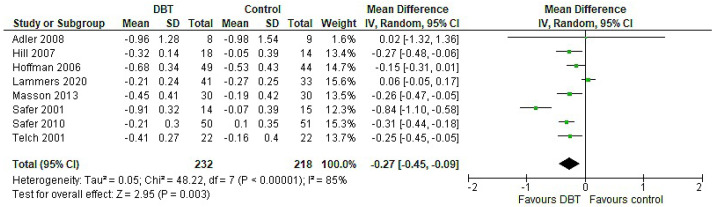
Forest plot: analysis of OBE.

**Figure 7 jpm-11-00931-f007:**
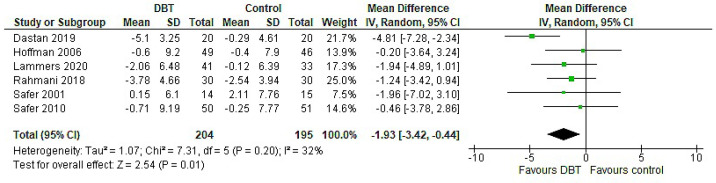
Forest plot: analysis of BMI.

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
