# Peer review of "Dialectical Behaviour Therapy Improves Emotion Dysregulation Mainly in Binge Eating Disorder and Bulimia Nervosa: A Systematic Review and Meta-Analysis"

_jpm, 2021, doi:10.3390/jpm11090931_

Round 1

Reviewer 1 Report

JPM-1350938 review

Introduction

  1. In the first paragraph, when you say ‘chronic’ and ‘long term’, can you be more specific about what time frame these research studies suggest? Are we suggesting across the duration of lifespan? Or just a much longer period. This distinction would be helpful.
  2. Also in the first paragraph, there are some minor typos – I won’t note these below, but please do a very close read to catch each of these.
  3. Second paragraph – you say ‘extrinsic’ and ‘intrinsic’ – what does this mean? Please use more precise language or cut this from this section.
  4. Please expand on IP’s interactive and social aspects – just a sentence will do.
  5. I am a bit underwhelmed with the Introduction’s discussion of DBT. Please carve out much more text on what it is and how it works. Also, at current, the discussion of DBT used in ED populations is cursory – please review a couple of these studies in detail in the Introduction.
  6. Please also add a sentence about the so-what factor – why do this meta? What will it add to the research literature?

Methods and Results

  1. How did you arrive at your date cut-off (i.e., 2000)?
  2. Please also add text as to why you included your exclusion criteria in relation to the effect probed in this meta.
  3. I am also curious why AN was included and if, in the Introduction when discussing DBT effects on EDs, how ER relates to AN versus binge eating based phenotypes (e.g., BED)? See above.
  4. Please expand on the secondary outcomes – give examples.
  5. Under 3.2, WL = wait list? Maybe remind the reader of this abbreviation, along with AT. I feel like some of these acronyms are a bit excessive and lead to unnecessary confusion in reading the paper. I suggest the authors streamline the main acronyms to aid in clarity throughout.
  6. Remove language about “suffered” or “suffering” – I suggest just saying “. . . of participants had BED” or the like.
  7. Overall, good, comprehensive discussion of results.

Discussion

  1. I appreciate your contextualization of results in terms of ED types (e.g., BED versus BN/AN). Still, I am looking for more information on what this means for future research. Part of this could be addressed in the Intro as noted above, but it should be repeated here that perhaps DBT used in these populations is in its infancy.
  2. Please expand on the link between EE and ER generally and its dimensions, especially given EE has been evidenced to prospectively predict disordered eating (namely BE). Furthermore, DBT has been extended as a useful framework in the treatment of EE – and even if just cursory, some discussion in this paper, especially if framed as future directions, could be useful. I suggest the authors examine the following, as well as their own search. Hopefully these articles help get more information (also suggest checking their Reference lists):
    1. Braden, A., Musher-Eizenman, D., Watford, T., & Emley, E. (2018). Eating when depressed, anxious, bored, or happy: Are emotional eating types associated with unique psychological and physical health correlates? Appetite, 125,410-417.
    2. Barnhart, W. R., Braden, A. L., & Price, E. (2021). Emotion regulation difficulties interact with negative, not positive, emotional eating to strengthen relationships with disordered eating: An exploratory study. Appetite, 158, 105038.
    3. Braden, A., Anderson, L., Redondo, R., Watford, T., Emley, E., & Ferrell, E. (2021)Emotion regulation mediates relationships between perceived childhood invalidation, emotional reactivity, and emotional eating. Journal of Health Psychology.
    4. Braden, A.& O’Brien, W. (2021). Pilot study of a novel treatment using dialectical behavioral therapy skills for adults with overweight/obesity and emotional eating. Journal of Contemporary Psychotherapy.
  3. Can you explain why you chose not to further examine asymmetry?
  4. I would also like the authors to create a Future Directions paragraph based on their findings to better guide readers in building on the meta.

Author Response

Dear Reviewer,

We would like to thank you for your effort in reviewing our submitted manuscript. Your comments and suggested modifications have helped us to improve the submitted manuscript significantly, for which we are grateful. Please find below our responses to the comments and details of the changes made to the manuscript.

Yours sincerely,

The Authors

Introduction

Comment 1: In the first paragraph, when you say ‘chronic’ and ‘long term’, can you be more specific about what time frame these research studies suggest? Are we suggesting the duration of lifespan? Or just a much longer period. This distinction would be helpful. 

The terms “chronic AN” and “chronic BN” refer to participants who maintained the eating disorder in studies with a 30- and 25- year follow up, respectively (Dobrescu et al., 2020; Steinhausen & Weber, 2009). With regards to the term “long term” BED participants were followed for 5 years (Clinical Practice Guidelines, 2009). These were clarified in the last two sentences of the first paragraph in introduction, as highlighted by the use of another font colour.

Comment 2:  Also in the first paragraph, there are some minor typos – I won’t note these below, but please do a very close read to catch each of these.

We have thoroughly read through the paragraph, as well as inserted the text into spelling check programs (several), without being able to identify any typos. We are open to this peer reviewer's suggestions for changes.

Comment 3: Second paragraph – you say ‘extrinsic’ and ‘intrinsic’ – what does this mean? Please use more precise language or cut this from this section. 

In this context, extrinsic refers to external influences on emotion regulation (typically family and friends), while intrinsic refers to internal, neurophysiological processes (e.g. development of cognitive skills). To make it clearer what we meant, we have changed intrinsic to internal and extrinsic to external. Changes made can be seen in the beginning of the second paragraph of the Introduction (section 1), as highlighted by the use of another font colour.

Comment 4: Please expand on IP’s interactive and social aspects – just a sentence will do. 

In the second paragraph of introduction, a sentence was added to briefly explain how IPT addresses  interaction and social problems in ED patients. 

Comment 5: I am a bit underwhelmed with the Introduction’s discussion of DBT. Please carve out much more text on what it is and how it works. Also, at current, the discussion of DBT used in ED populations is cursory – please review a couple of these studies in detail in the Introduction. 

We agree with this peer reviewer and have added  two new paragraphs in the introduction, beginning and ending as follows (also marked by the different font colour):

“DBT is based on...and distress tolerance” & “Since the biosocial...frequency of ED symptoms”.  

Comment 6: Please also add a sentence about the so-what factor – why do this meta? What will it add to the research literature? 

We agree that it is essential to provide the reasoning behind the research conducted, as well as the impact it can have on the current state of evidence. While the ending of the introduction provides some suggestions, we acknowledge that direct mention is more suitable. Therefore, we created a separate paragraph to capture the ideas more clearly.  Changes made can be seen in the new third paragraph of the Introduction (section 1), as highlighted by the use of another font colour.

Methods and Results 

Comment 7: How did you arrive at your date cut-off (i.e., 2000)?  

In section 2.2. two sentences were added explaining the rationale behind choosing the date cutoff of 2000. Because DBT was examined as a treatment for ED after the year of 2000, and therefore the knowledge on this topic became clearer, we decided to search for studies published  in the last 20 years. However, papers published earlier than 2000, found in various sources, were still considered in this study. 

Comment 8: Please also add text as to why you included your exclusion criteria in relation to the effect probed in this meta. 

We have added corresponding reasons for exclusion following every exclusion criteria. Changes made can be seen in 2.2, as highlighted by the use of another font colour.

Comment 9: I am also curious why AN was included and if, in the Introduction when discussing DBT effects on EDs, how ER relates to AN versus binge eating based phenotypes (e.g., BED)? See above.  

AN was included according to the aim of the study to investigate the effect of DBT in all ED. However, the literature search revealed scarce data on AN and we have emphasized in our conclusion the need to increase research in this specific population.  

Comment 10: Please expand on the secondary outcomes – give examples.  

The secondary outcomes were further explained in section 2.3. by adding the specific Outcome Measure Instrument (OMIs) used to measure them and describing what they assess. 

Comment 11: Under 3.2, WL = wait list? Maybe remind the reader of this abbreviation, along with AT. I feel like some of these acronyms are a bit excessive and lead to unnecessary confusion in reading the paper. I suggest the authors streamline the main acronyms to aid in clarity throughout.  

The abbreviations “AT” and “WL” were spelled out throughout the entire paper. In addition, “ER” was replaced with “Emotion Regulation (ER)” in the beginning of several paragraphs to remind readers the meaning of the abbreviation. 

Comment 12: Remove language about “suffered” or “suffering” – I suggest just saying “. . . of participants had BED” or the like.  

The search function was used to identify the terms, which have been removed and replaced as requested.

Discussion

Comment 14: I appreciate your contextualization of results in terms of ED types (e.g., BED versus BN/AN). Still, I am looking for more information on what this means for future research. Part of this could be addressed in the Intro as noted above, but it should be repeated here that perhaps DBT used in these populations is in its infancy. 

After mentioning the lack of BN and AN participants in the current study, we have added a sentence explaining that research is still in its infancy for these types of ED (In section 4.1. highlighted by the use of another font colour).

Comment 15: Please expand on the link between EE and ER generally and its dimensions, especially given EE has been evidenced to prospectively predict disordered eating (namely BE). Furthermore, DBT has been extended as a useful framework in the treatment of EE – and even if just cursory, some discussion in this paper, especially if framed as future directions, could be useful. I suggest the authors examine the following, as well as their own search. Hopefully these articles help get more information (also suggest checking their Reference lists).  

We have examined the suggested literature, and therefore have emphasized the link between the specific dimensions of both EE and ER. Directions for future studies were also provided, and changes can be seen in the second paragraph of section 4.1., as highlighted by the use of another font colour.

Thus, when individuals experience both negative emotional eating and poor emotion regulation, risk for  experiencing concerns about weight and a range of disordered eating symptoms may be higher. Furthermore, results suggest that emotion regulation may be a  viable treatment target in  concurrent presentations of  negative emotional eating and DE such that improving emotion regulation may reduce risk for  experiencing weight concerns and other disordered eating symptoms among individuals with concurrent negative emotional eating.

Comment 16: Can you explain why you chose not to further examine asymmetry? 

We chose not to examine asymmetry, as we considered the risk of publication bias to be low, based on the clear symmetry observed in the funnel plots.

Comment 17: I would also like the authors to create a Future Directions paragraph based on their findings to better guide readers in building on the meta.

A new section has been added at the end of discussion (Section 4.6) which recommends three directions for future research.

Reviewer 2 Report

JPM-1350938 Reviewer Comments

Title: Dialectical *Behaviour Therapy Improves Emotion Dysregulation Mainly in Binge Eating Disorder and Bulimia Nervosa: A Systematic Review and Meta-Analysis

Reviewer Comments:

Summary of Manuscript:

This systematic review and meta-analysis investigated whether the effect of Dialectical Behavior Therapy (DBT) was stronger on emotion dysregulation, general psychopathology, and Body Mass Index (BMI), in patients with Eating Disorders (ED), when compared to a control group.

Eleven studies were identified in a systematic search in accordance with PRISMA guidelines. Most studies included patients with Binge Eating Disorder (BED) (n = 8), some with Bulimia Nervosa (BN) (n = 3) and only one with Anorexia Nervosa (AN). The pooled effect of DBT indicated a greater improvement in Emotion Regulation (ER) (g = -0.69, p = 0.01), depressive symptoms (g = -0.33, p<0.00001), ED psychopathology (MD = -0.90, p = 0.005), Objective Binge Episodes (OBE) (MD = -0.27, p = 0.003), and BMI (MD = -1.93, p = 0.01), compared to the control group. No improvement was detected in eating ER following DBT (p = 0.41). The authors conclude that DBT demonstrated greater efficacy than the control group in improving emotion dysregulation, ED psychopathology, and BMI, in ED. Noted study limitations included the small number of studies, and high variability in between studies.

*Given the original spelling of Dialectical *Behavior Therapy (DBT), is it acceptable to the journal to use the modified spelling of *Behaviour (which may be more common in Denmark)?

Overall Impression:

This manuscript adds to the literature by providing the first systematic review and meta-analysis exclusively investigating the effect of DBT on ER in relation to a control group in all ED. Additionally, measures of general psychopathology and BMI also contribute to the findings.

Please see below for specific comments in each area.

*Title:

*See my note above re: *Behaviour vs *Behavior.

The current title is detailed and accurate to the content of the manuscript.

Abstract:

Overall, the content of the abstract is a nice summary of the systematic review and meta-analysis. However, please consistently use the term “participants” vs “patients” or “subjects” here and throughout the manuscript. Please also clarify what you mean by “control group” here (active therapy [AT] or waitlist [WL]) and throughout the manuscript.

Keywords:

*See my note above re: *Behaviour vs *Behavior.

The current keywords are accurate (emotion regulation; eating disorders; dialectical *behaviour therapy), though since the majority of the studies included in the systematic review and meta-analyses include participants with BED and BN, perhaps you can include these keywords also.

Introduction:

The authors provide a succinct overview of the literature with excellent references, leading up to their rationale for the current systematic review and meta-analysis.

Materials and Methods:

*See my note below re: bold font the mention of all tables, figures, and supplementary materials.

Overall, the materials and methods section is outlined very thoughtfully. For the study exclusion criteria, what is the rationale for excluding participants below 12 years old (e.g., DBT is not an EBT for this younger age group)? For the outcomes, as an augment to the excellent abbreviation notations for multiple assessments in Tables 1 and 2, it would be nice to provide a key to link the different OMIs to the specific variables that they are measuring/you are including in your meta-analysis. This may improve clarity for the reader who may not be able to correspond the text overview to the Table listings of measurements. For example, in the text, does the “eating ER (OMI depicting desire to eat in response to negative affect)” refer to the Emotional Eating Scale (EES) with the global and subscales (e.g., anger/frustration, anxiety, depression) in the Tables?

Results:

The presentation of the results is well organized. The included study characteristics highlight the limitations of the current literature (e.g., small number of eligible studies, small Ns, short follow-up assessment periods, mostly conducted in the USA, predominantly female BED participants). Additionally, the overall quality of the current review was rated as intermediate (3 studies at low risk of bias, 4 unclear, and 3 high risk). While the authors are obviously not responsible for the quality of the overall data from the original studies, they do an excellent job of detailing the ways in which the studies are flawed/can be improved upon in the future.

That being said, the meta-analysis does provide promising initial results that DBT can improve ER, depressive symptoms, ED psychopathology, OBE, and BMI in ED compared to controls. The finding that there was no improvement in eating ER following DBT is somewhat less encouraging, though the authors provide a good discussion of some factors that may influence this, including the lack of longitudinal data to demonstrate a pattern of improvement over time.

Discussion:

The authors give a well-outlined and thoughtful discussion of the results. For the Strengths and Limitations subsection, is it acceptable to the journal to have this within the Discussion section? For other journals, they specifically request a separate section (i.e., Discussion, Limitations, Conclusions) in order to allow a thorough exploration of the factors that limit generalizability.

Conclusions:

The conclusions section provides a nice summary of the results, with explicit attention paid to the limitations of the included studies. The final sentence, “Additionally, it would be pertinent to analyze individual factors which can contribute to the development of specific ED interventions” is interesting, and depending upon the journal/authors, may be worth adding to the discussion.

Tables/Figures/Supplementary Material:

*Please bold font the mentions of all tables, figures, and supplementary materials in the text so that the reader can more clearly see and refer to them.

Back Matter: All other sections of the manuscript seem to be provided per JPM:

Data Availability Statement

Author contributions

Funding

Acknowledgements

Conflicts of interest

Abbreviations

Appendixes

References

Supplementary material

Author Response

Dear Reviewer,

We would like to thank you for your effort in reviewing our submitted manuscript. Your comments and suggested modifications have helped us to improve the submitted manuscript significantly, for which we are grateful. Please find below our responses to the comments and details of the changes made to the manuscript.

Yours sincerely,

The Authors

Summary

Comment 1: *Given the original spelling of Dialectical *Behavior Therapy (DBT), is it acceptable to the journal to use the modified spelling of *Behaviour (which may be more common in Denmark)?

Although we understand that the original spelling was *Behavior, we have followed the JPM instructions for authors where it is mentioned that consistency should be maintained regardless of the spelling style chosen, which in our case was UK English. See below.

 “American English or UK English are fine so long as there is consistency.”

Abstract

Comment 2: Overall, the content of the abstract is a nice summary of the systematic review and meta analysis. However, please consistently use the term “participants” vs “patients” or “subjects”  here and throughout the manuscript. Please also clarify what you mean by “control group” here  (active therapy [AT] or waitlist [WL]) and throughout the manuscript. 

All terms were changed to “participants” when referring to a study that was carried out, and “patients” for more general referrals to individuals with an ED. Active therapy and wait list were added throughout the manuscript to define the control group.

Keywords

Comment 3: *See my note above re: *Behaviour vs *Behavior. The current keywords are accurate (emotion regulation; eating disorders; dialectical *behaviour therapy), though since the majority of the studies included in the systematic review and meta analyses include participants with BED and BN, perhaps you can include these keywords also.

Retrospectively, we agree that BED and BN can be important keywords, and therefore we have included both of them in our list. 

Materials and Methods

Comment 4: *See my note below re: bold font the mention of all tables, figures, and supplementary materials. Overall, the materials and methods section is outlined very thoughtfully. For the study exclusion  criteria, what is the rationale for excluding participants below 12 years old (e.g., DBT is not an  EBT for this younger age group)? For the outcomes, as an augment to the excellent abbreviation  notations for multiple assessments in Tables 1 and 2, it would be nice to provide a key to link the  different OMIs to the specific variables that they are measuring/you are including in your meta analysis. This may improve clarity for the reader who may not be able to correspond the text  overview to the Table listings of measurements. For example, in the text, does the “eating ER  (OMI depicting desire to eat in response to negative affect)” refer to the Emotional Eating Scale  (EES) with the global and subscales (e.g., anger/frustration, anxiety, depression) in the Tables?

We agree that bolding the mentions of tables, figures and supplementary material can help better reference them, and therefore have added this throughout the manuscript. Further, the rationale for exclusion criteria can also be of great importance, and for this reason it was provided in section 2.2., along with the reasoning behind the age cut-off. Lastly, in section 2.3., the OMIs were specified as an attempt to provide clarity for the reader in text. 

Discussion

Comment 5: The authors give a well-outlined and thoughtful discussion of the results. For the Strengths and  Limitations subsection, is it acceptable to the journal to have this within the Discussion section?  For other journals, they specifically request a separate section (i.e., Discussion, Limitations, Conclusions) in order to allow a thorough exploration of the factors that limit generalizability.

We have incorporated the Strengths and Limitations into the Discussion section, as suggested by the Instructions for Authors provided by Journal of Personalized Medicine. See below.

“Authors should discuss the results and how they can be interpreted in perspective of previous studies and of the working hypotheses. The findings and their implications should be discussed in the broadest context possible and limitations of the work highlighted. Future research directions may also be mentioned. This section may be combined with Results.”

Conclusion

Comment 6: The final sentence, “Additionally, it would be pertinent to  analyze individual factors which can contribute to the development of specific ED interventions”  is interesting, and depending upon the journal/authors, may be worth adding to the discussion. 

We agree that it is an interesting topic and we consider that we have touched upon that to some extent in the second paragraph of section 4.5.1., as well as in section 4.6. We have also added specific examples in the conclusion, to clarify what we were referring to.final

Round 2

Reviewer 1 Report

Good work! I am satisfied with the authors' integration of my comments, and commend them for an important meta/review!